# Chaotic Characteristic Analysis of Vibration Response of Pumping Station Pipeline Using Improved Variational Mode Decomposition Method

**Li Jiang [1,2], Zhenyue Ma [1], Jianwei Zhang [2,*], Mohd Yawar Ali Khan [3], Mengran Cheng [2] and Libin Wang [2]**

[1]  School of Hydraulic Engineering, Dalian University of Technology, Dalian 116024, China; jiangli@ncwu.edu.cn (L.J.); dmzy@dlut.edu.cn (Z.M.)

[2]  School of Water Conservancy, North China University of Water Resources and Electric Power, Zhengzhou 450046, China; cmr6688@126.com (M.C.); wanglb1994@126.com (L.W.)

[3]  Department of Hydrogeology, King Abdulaziz University, Jeddah 21589, Saudi Arabia; makhan7@kau.edu.sa

*  Correspondence: zjwcivil@126.com

**Abstract:** The measured vibrational responses of the pumping station pipeline in the irrigation site were chosen to confirm the chaotic characteristics of the pumping station pipeline vibration and to determine the vibrational excitation that makes it chaotic. First, the chaotic properties of the pipeline vibration responses were investigated using a saturation correlation dimension and the maximum Lyapunov exponent. The vibration excitation with chaotic features was obtained using an improved variational mode decomposition (IVMD) method to examine the multi-time-scale chaotic characteristics of the pipeline vibration responses. The results show that the vibrational responses of each measuring point of the pipeline under different operating conditions have clear chaotic characteristics, where the chaotic characteristics of the axial points and bifurcated pipe points are relatively strong. The vibration of the operating conditions and measurement points affected by the unit's operation and flow state change is further complicated. The intrinsic mode function (IMF) produces a low-dimensional chaotic attractor after the IVMD disrupts the vibration response. Still, the vibration excitation of the remaining components on behalf of the units does not have chaotic properties, implying that water pulsation excitation makes the pumping station pipeline vibrations chaotic. The vibration excitation caused by the unit's operation covers the chaotic characteristics of the pipeline vibration and increases its uncertainty. The outcomes of this study provide a theoretical basis for further exploration of the vibration characteristics of pumping station pipelines, and a new method of chaos analysis is proposed.

**Keywords:** pumping station pipeline; chaotic characteristic; IVMD; vibration response; correlation dimension; Lyapunov exponent



## 1. Introduction

High-lift pumping stations and water-diversion irrigation areas have been built in many water-deficient areas due to the continuous development of electric water-lifting equipment and water-diversion irrigation technology in China. These projects have created enormous economic, ecological and social benefits. Thus, ensuring their safe and stable operation is the main task of modernising and developing water conservation in China [1]. Natural and human forces create varying degrees of pipeline vibration during long-term operation at pumping stations [2]. Long-term irregular pipe vibration will lead to the loosening of the pipelines and their auxiliary system, causing catastrophic damage in severe cases [3]. Therefore, it is of great research interest to analyse the vibration characteristics of the pumping station pipeline to avoid its adverse vibrations.

Chaos is a unique mechanical phenomenon in the vibration of strongly nonlinear structures. Most researchers believe that the vibrations of pipelines are weakly nonlinear,

so they mainly focus on studying chaotic phenomena caused by the flow of water and other excitation sources such as flow. There is a scarcity of research on the chaotic processes in the pipeline itself when researching nonlinear problems. It is found that the chaotic phenomena of pipelines do not depend solely on the strength of structural nonlinearity; for some weak nonlinear or even linear structures, chaos occurs [4]. PaïDoussis studied the dynamics of a cantilever pipeline with nonlinear constraints and constant internal flow, which discovered the chaotic motion of the system [5]. Tang obtained the chaotic characteristics of the transport pipeline by increasing the nonlinear force and found that the occurrence of chaos is mainly affected by the flow velocity in the pipeline [6]. B.G. Sinir investigated the nonlinear vibrations of slightly curved pipes that transport fluid with constant velocity [7]. The periodic and chaotic movements have been observed in the transverse vibrations of slightly curved pipes transporting fluid. Zhao analysed the chaotic phenomenon in the pipeline vibration caused by the flow pulsation excitation under thermal load and then obtained the relationship between the frequency response and flow velocity [8].

Research on the chaotic characteristics of pipeline systems mainly focuses on oil-gas pipelines and the mathematical models of pipelines with specific nonlinear constraints. In contrast, the chaotic characteristics of pumping station pipeline systems are rarely studied. Most of the previous research achievements have only analysed the chaotic characteristics of the vibration system but have not further explored the vibrational excitation that caused the chaos. In this paper, the measured vibration responses of the pumping station pipeline in an irrigation area are taken as the research objective. The chaotic characteristics of the vibratory responses of the pumping station pipeline under different working conditions are analysed by using the saturation correlation dimension and the largest Lyapunov exponent. In addition, the IVMD method is used to decompose the vibration responses of the measurement points under typical working conditions. The chaotic characteristics of the IMFs are analysed to obtain the vibratory excitation that causes the chaotic characteristics of the pumping station pipeline.

## 2. Theoretical Aspects

### 2.1. Identification Method of Chaotic Characteristics

There are many methods for identifying chaotic characteristics, which are roughly divided into qualitative and quantitative analysis. Orbit observation, Poincare surface of section, and power spectral analysis are examples of qualitative approaches [9–11]. These methods are feasible and straightforward, but they are limited in determining whether the system has chaotic characteristics and cannot perform transverse comparisons under different operating conditions. Quantitative methods, such as the saturation correlation dimension method [12] and the largest Lyapunov exponent method [13], can reflect the vibration complexity and the degree of chaos under different conditions by comparing the values of the parameters. The saturation correlation dimension and the largest Lyapunov exponent are chosen as the chaotic identification indexes of pipeline vibration responses in pumping stations to improve the trustworthiness of the result.

#### 2.1.1. Saturation Correlation Dimension

The correlation dimension characterises the compactness of a dynamic system and is used to reflect the system's complexity. When the saturation correlation dimension is fractional, the system is said to have chaotic properties. For an m-dimensional phase space, its correlation function can be defined as follows:

$$C(r) = \lim_{M \to \infty} \frac{2}{M(M-1)} \sum_{1 \le i \le j \le M} H(r - \|Y_i - Y_j\|) \tag{1}$$

where $M = N - (m-1)\tau$ is the number of phase points, $H(u)$ is the Heaviside function, $N$ is the time series; $r$ is the vector point in the time series; $M$ is the embedding dimension; Tau is the time delay; $Y$ is the reconstruction vector.

When the time series is chaotic, for the positive, the relationship between the correlation function $C(r)$ and $r$ is

$$C(r) \propto \alpha r^{D_2} \tag{2}$$

where $\alpha$ is a constant, $D_2$ is the correlation dimension which can be obtained by the slope of the $\log_2 C(r) \sim \log_2 r$ curve, that is

$$D_2 = \lim_{r \to 0} \frac{\log_2 C(r)}{\log_2 r} \tag{3}$$

Due to the noise in the measured signal, the embedding dimension is generally controlled to rise gradually. The apparent straight line segments in the $\log_2 C(r) \sim \log_2 r$ curve are fitted using the least square method for each embedding dimension. The slope of each segment increases with the rising embedding dimension, and eventually reaches saturation, the saturation correlation dimension.

### 2.1.2. Largest Lyapunov Exponent

The Lyapunov exponent determines the chaotic characteristics of the system based on the diffusion of the phase trajectory. Generally, the direction represented by the positive Lyapunov exponent supports the attractors. In contrast, the contraction direction corresponding to the negative Lyapunov exponent contributes to the attractor dimension's fractional part after counteracting the expansion direction's effect. Thus, the positive Lyapunov exponent is a prominent feature of chaos. Suppose $\lambda_1$ as the largest Lyapunov exponent of a system, then the chaotic components of the system can be found if $\lambda_1$ is positive, and its value reflects the chaos degree.

Rosenstein [14] proposed the small data sets for computing $\lambda_1$. Its basic steps are as follows:

Choose suitable $\tau$ and $m$ to reconstruct the phase space and find the nearest neighbor point $Y_{\hat{i}}$ of each $Y_i$ in the phase space. Short separation limitation is as follows:

$$d_i(0) = \min_{\hat{i}} \|Y_i - Y_{\hat{i}}\| (|i - \hat{i}| > p) \tag{4}$$

where $p$ is the average period of time series, $i$ is the vector in space, $\hat{i}$ is the vector of the nearest neighbor of the second vector.

Define the distance of $Y_{i+j}$ and $Y_{\hat{i}+j}$ as

$$d_i(j) = \|Y_{i+j} - Y_{\hat{i}+j}\| \tag{5}$$

where $j = 0, 1, 2, \cdots, \min(M - i, M - \hat{i})$.

For each $j$, compute the $\ln d_i(j)$ average as follows:

$$y(i) = \frac{1}{q \Delta t} \sum_{i=1}^{q} \ln d_i(j) \tag{6}$$

where $q$ is the number of nonzero $\ln d_i(j)$. The slope of the regression line made by the least square method is $\lambda_1$.

### 2.2. Improved Variational Mode Decomposition (IVMD)

Variational mode decomposition (VMD) is a new method of multi-component adaptive signal decomposition [15]. Compared to traditional signal decomposition methods, it effectively avoids modal aliasing and over-decomposition defects and has a higher utilization value [16]. VMD comprises two processes comprising the establishment of variational constraints and iteration to find the optimal solution. The specific operation process is as follows: VMD decomposes a given signal $f$ into $K$ modal functions using variational constraints $m_k(t)$. The bandwidth of each IMF is limited, and each IMF is distributed around the central pulsating frequency. The variational constraint model is as follows [17]:

$$\begin{cases} \min\limits_{m_k, w_k} \left\{ \sum\limits_k \left\| \partial_t \left[ \left( \sigma(t) + \frac{j}{\pi t} \right) m_k(t) \right] e^{-jw_k t} \right\|_2^2 \right\} \\ s.t. \sum\limits_k m_k = f \end{cases} \quad (7)$$

where $\{m_k\}$ represents the decomposed K IMF components, $\{m_k\} = \{m_1, m_2, \cdots, m_k\}$; $\sigma(t)$ is a pulse function; $\{w_k\}$ is the central frequency of each IMF, $\{w_k\} = \{w_1, \ldots w_k\}$.

To complete the adaptive decomposition of input signals $f$ and to obtain the IMFs with the minimum sum of bandwidth, the following expanded Lagrange expression is introduced:

$$L(m_k, w_k, \lambda) = \alpha \sum_k \left\| \partial(t) \left[ \left( \delta(t) + \frac{j}{\pi t} \right) m_k(t) \right] e^{-jw_k t} \right\|_2^2 + \left\| f(t) - \sum_k m_k(t) \right\|_2^2 + \left\langle \lambda(t), f(t) - \sum_k m_k(t) \right\rangle \quad (8)$$

where $\alpha$ is the penalty factor to ensure the accuracy of signal reconstruction; $\lambda(t)$ is a Lagrange multiplier used to strengthen the constraint; $\langle \rangle$ represents the inner product operation.

To solve the above variational constraint problem, the dual decomposition and alternate direction multiplication sub-algorithm are used [18]. Keep updating $m_k$, $w_k$ and $\lambda(t)$ to find the saddle point of Equation (8), that is, the optimal solution of Equation (7). The modal component function $m_k$ and the central frequency $w_k$ are

$$m_k^{n+1}(w) = \frac{f(w) - \sum\limits_{i \neq k} m_i(w) + \frac{\lambda(\omega)}{2}}{1 + 2\alpha(w - w_k)^2} \quad (9)$$

$$w_k^{n+1} = \frac{\int_0^\infty w |m_k(w)|^2 dw}{\int_0^\infty |m_k(w)|^2 dw} \quad (10)$$

$$\lambda^{n+1} = \lambda^n + \tau \left( f(w) - \sum_k m_k^{n+1}(w) \right) \quad (11)$$

When VMD decomposes the vibration response sequence, determining the total modal number is a crucial step. The selection of modal parameters $K$ greatly affects the accuracy of the results [19]. A parameter $K$ is usually challenging to determine. If $K$ is greater than the number of useful components obtained by signal decomposition, information superposition will occur; if $K$ is smaller than it, a part of the limited bandwidth of the solid modulus cannot be decomposed. An IVMD method based on the mutual information (MI) method is proposed for $K$ selection.

MI reflects the correlation between two random variables and allows better identification of the degree of correlation [15]. MI is as follows:

$$I(X, Y) = H(Y) - H(X|Y) \quad (12)$$

where $H(Y)$ is the entropy of $Y$, and $H(Y|X)$ is the conditional entropy of $Y$ when $X$ is known. When $I(X, Y) = 0$, $X$ and $Y$ are independent of each other.

The mutual information $I_k$ of the original signal and each IMF obtained by the IVMD decomposition is calculated and normalised by Equation (13). Then the correlation between each modal component and the original signal is judged, that is, whether the original signal is completely decomposed.

$$\sigma_i = \frac{I_i}{\max(I_i)} \quad (13)$$

where $\sigma_i$ is the normalized mutual information value of each IMF, i $=1, 2, \ldots k$. Refer to reference [20], when $\sigma_i$ is less than 0.02, it is considered that the IMF does not contain valid feature information. The original signal has been decomposed completely.

The specific algorithm for adaptive determination of K using MI method is as follows:

Step 1: Initialize $n = n + 1$, assign $K = 1$;

Step 2: $K = K + 1$, perform outer circulation;

Step 3: Initialize $m_k^1, w_k^1, \lambda^1$ and $n$, assign $n = 0$;

Step 4: Order $n = n + 1$ to execute the inner loop;

Step 5: For all $w \geq 0$, according to Equations (9) and (10), $m_k$ and $w_k$ are updated, respectively;

Step 6: Update $\lambda$ according to Equation (11);

Step 7: For a given discriminate accuracy $e > 0$, if the iteration condition $\sum_k \frac{\|m_k^{n+1} - m_k^n\|_2^2}{\|m_k^n\|_2^2} < e$ is satisfied, the process is terminated, otherwise loop step 2 to step 6;

Step 8: Circulate step 2 to 7 until the set threshold $\sigma$ is greater than the normalized mutual information $\sigma_i$, that is, if $I(f - \sum m_k, f) < \sigma$, end the cycle.

The flow chart of the above calculation steps is shown in Figure 1.

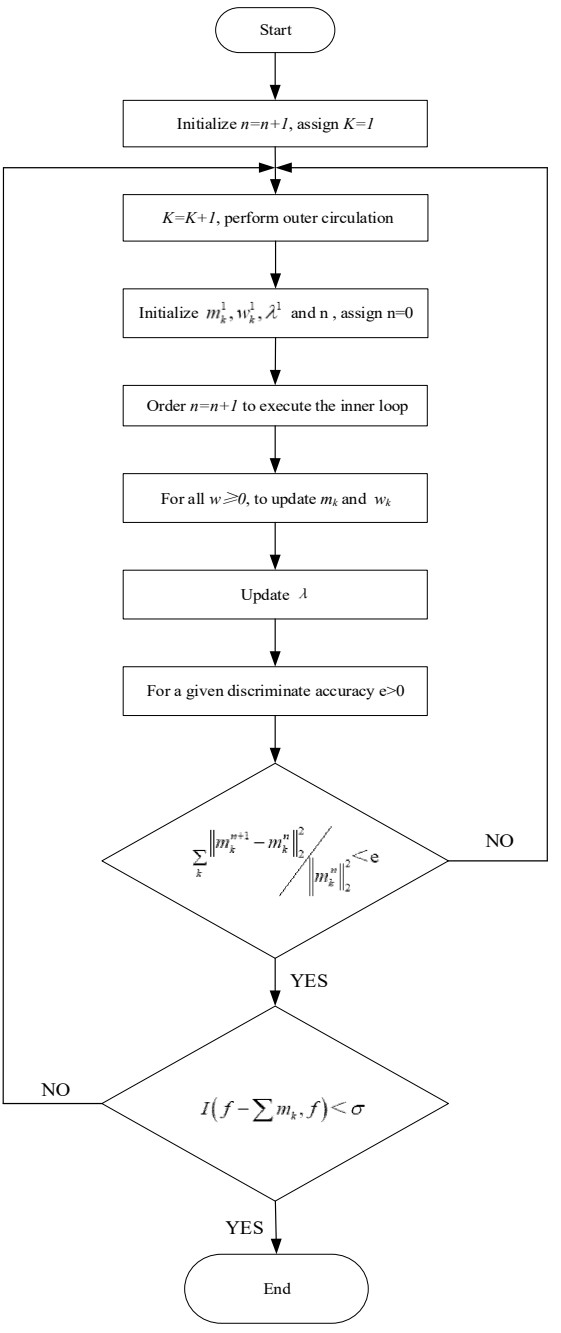

**Figure 1.** Flow chart for adaptive determination of K.

## 3. Chaotic Characteristics Analysis of Pipeline Vibration Response

The pipe material is stainless steel. Model 891–2 vibration sensors are used in the test, which are divided into four grades: small speed, medium speed, large speed and acceleration. So speed sensors are used in this test. Taking the No. 2 pressure pipeline of the Jingdian Project pumping station No. 3 as a research objective, the No. 4 and No. 5 units of a 1200S–56 horizontal centrifugal pump are connected with the branch pipe. Six measurement points are selected on the main pipe and two branches of the pipeline. Each point is equipped with vibration sensors in X, Y and Z directions. The measuring points are arranged as shown in Figure 2.

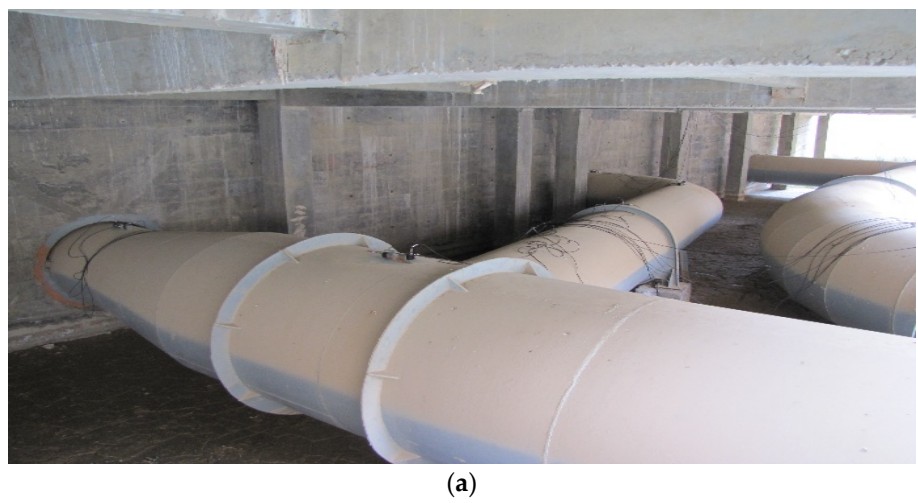

(**a**)

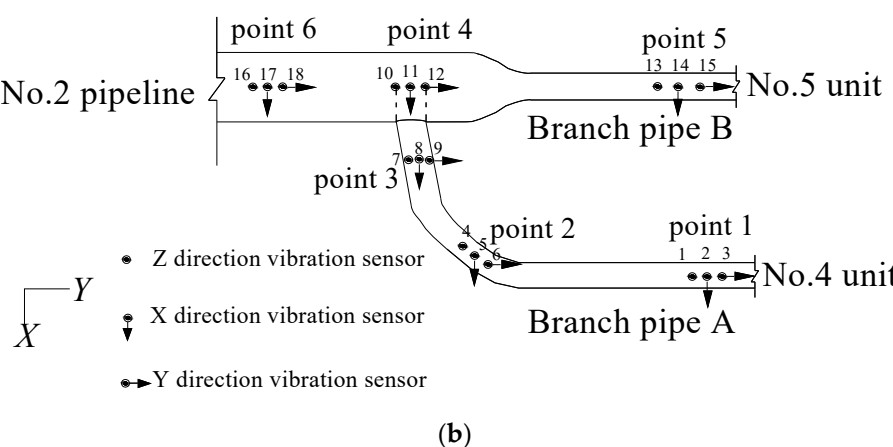

(**b**)

**Figure 2.** Layout of pipeline measuring points (**a**) Field test of pipeline, and (**b**) Measuring points layout (Note: 1~18 is the sensor number).

In the prototype test, four working conditions were selected to collect the vibration responses of the pipeline. The descriptions of each working condition, sampling time and sampling frequency are shown in Table 1.

**Table 1.** Four working conditions.

| Cases | Description of Working Conditions | Sampling Time/s | Sampling Frequency/Hz |
|---|---|---|---|
| 1 | No. 4 unit stable operating | 900 | 512 |
| 2 | No. 4 unit opening | 1800 | 512 |
| 3 | No. 4 unit closing | 1800 | 512 |
| 4 | No. 4 and 5 units stable operating | 900 | 180 |

The velocity-time history of points under typical conditions is shown in Figure 3. The chaotic characteristic analysis of the vibration responses under different conditions is carried out as follows:

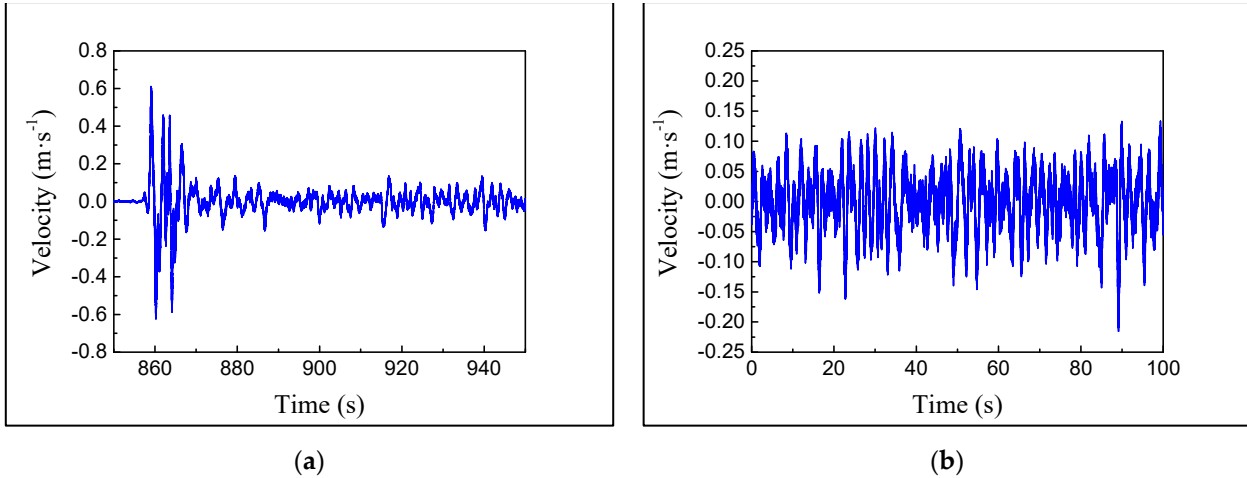

(**a**)                                                                                     (**b**)

**Figure 3.** Velocity time history of points under typical conditions (**a**) Z-axis vibration of point 1 under condition 2, and (**b**) Z-axis vibration of point 1 under condition 4.

First, the reconstruction of the phase space of the time series is performed, that is, the calculation of the time delay $\tau$ and embedding dimension $m$. The CAO method essentially uses the minimum error method to determine the embedding dimension, which was proposed by Liangyue Cao in 1997. This paper calculates $\tau$ by the autocorrelation function method and chooses the CAO method to obtain $m$ [21]. The calculation process of $\tau$ and $m$ is illustrated by taking the Z-axis vibration of point 1 under condition 4 as an example.

In the process of calculating $\tau$ by the autocorrelation function method, when the value drops to $1-1/e$ of the initial value, the corresponding time delay is $\tau$. The result of the autocorrelation function is shown in Figure 4.

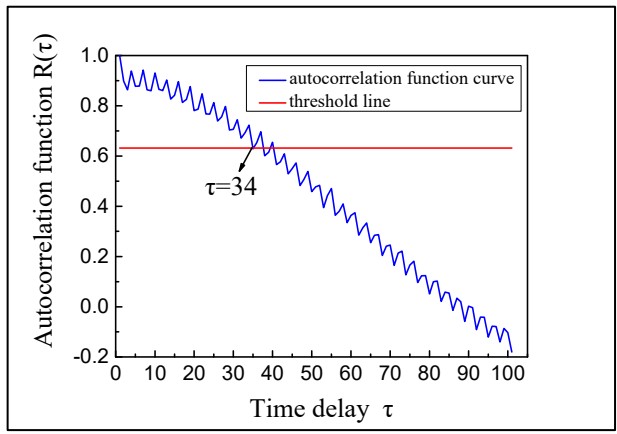

**Figure 4.** $\tau$ Calculation of point 1 Z-axis vibration under condition 4.

After obtaining $\tau$, the embedding dimension is determined by the CAO method. $E_1(m)$ represents the minimum embedding dimension. $E_2(m)$ represents the characteristics of time series. When $E_1(m)$ obviously no longer changes with the increase, and the $E_2(m)$ value tends towards 1, the corresponding $m$ is the optimal embedding dimension. From Figure 5, we can see that the optimal embedding dimension $m$ of point 1 Z-axis vibration under condition 4 is 11.

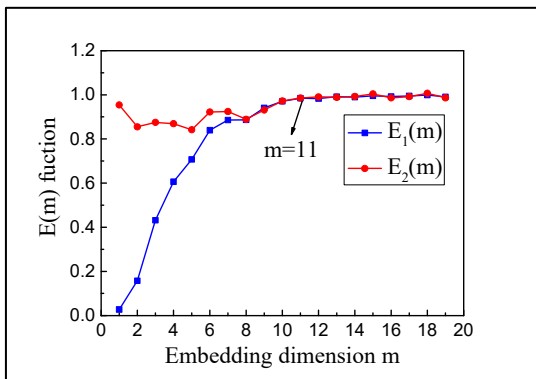

**Figure 5.** *m* Calculation of point 1 Z-axis vibration under condition 4.

The G-P algorithm [22] and the small data sets are chosen to calculate the saturation correlation dimension and the largest Lyapunov exponent. Two types of indexes are used to analyse the chaotic characteristics of time series.

The G-P algorithm is a chaotic eigenvalue calculation method proposed by Grassberger and Procaccia to calculate the saturation correlation dimension $D_2$.

The embedding dimension is selected as $m = 2, 4, 6, \cdots, 20$ and the $\tau$ has been calculated above. According to the correlation function relation in Equation (3), the $\log_2 C(r) \sim \log_2 r$ double logarithmic relation graph of different $m$ is plotted, respectively. The slope fitted by the near line segment of the curve is the correlation dimension under the corresponding embedding dimension. As the embedding dimension increases, it is the saturation dimension $D_2$ when the correlation dimension reaches saturation. Figure 6 is the diagram representing the calculation of the saturation correlation dimension of specific points.

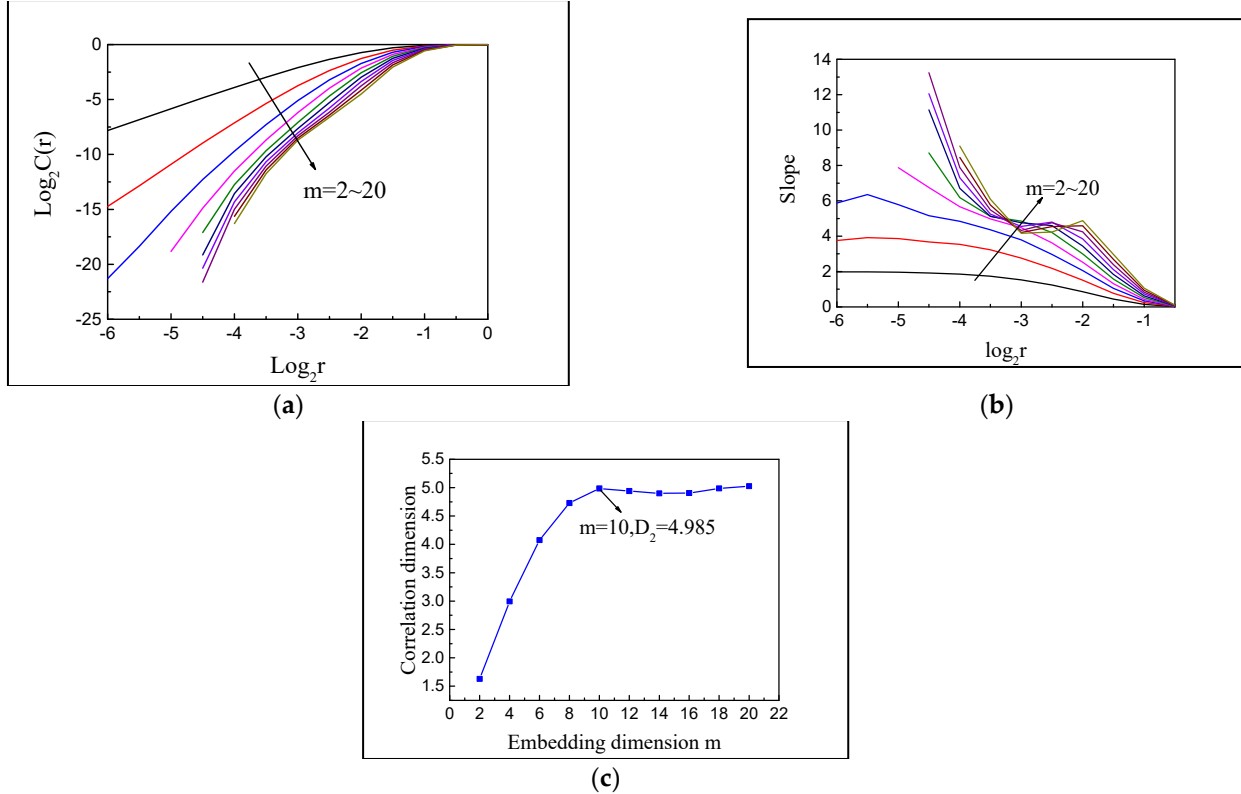

**Figure 6.** Calculation of point 1 z-axis vibration under condition 4 (**a**) Double logarithmic curve, (**b**) Slope of double logarithmic curve, and (**c**) Relation between $D_2$ and m.

To reveal the distribution law of the saturation correlation dimension, the $D_2$ variation curves of points in each direction under different conditions are shown in Figure 7.

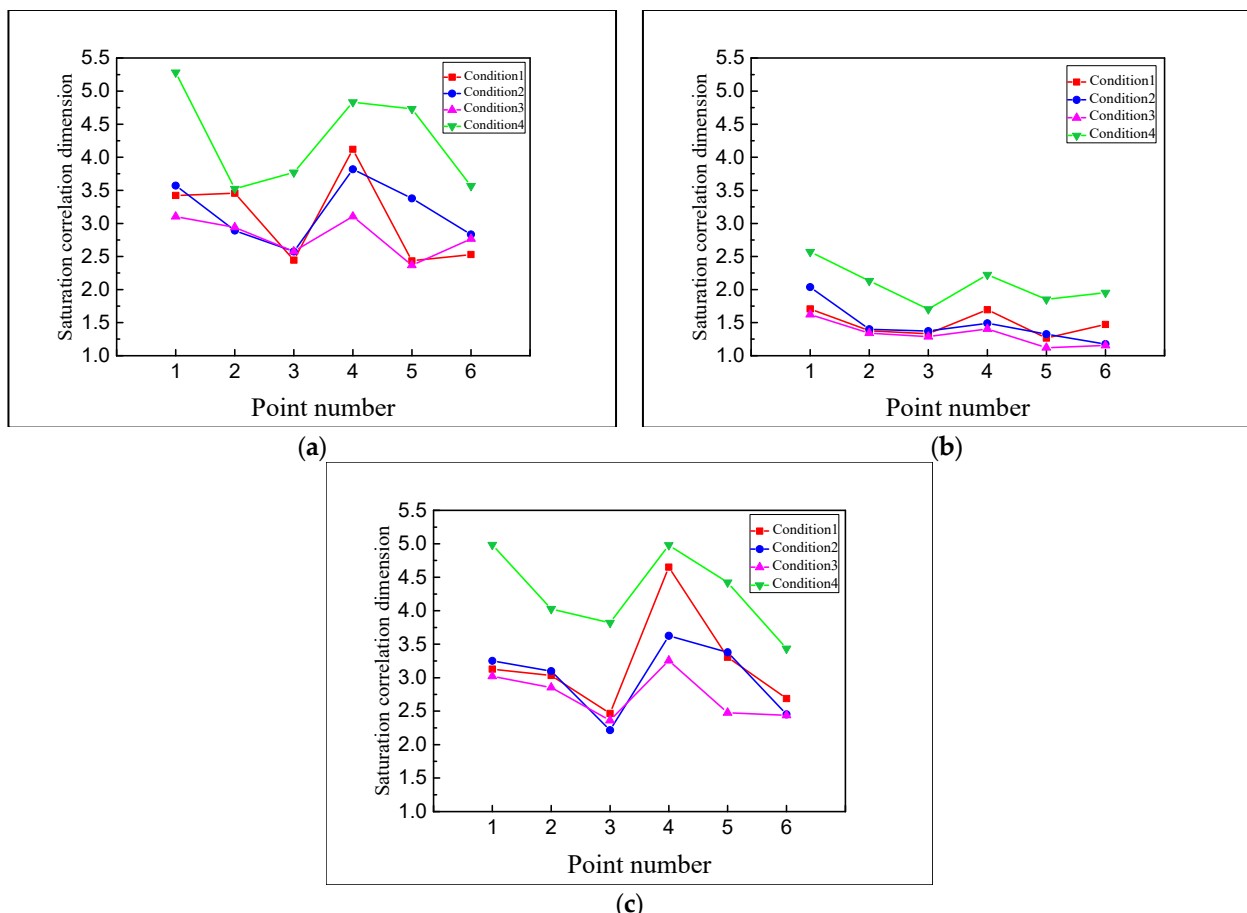

**Figure 7.** Correlation dimension curves of points in different directions (**a**) X-axis points, (**b**) Y-axis points, and (**c**) Z-axis points.

As can be seen from Figure 7:

(1) In general, $D_2$ ranges from 1.156 to 5.283, and they are fractional, indicating that the responses of the pipeline in all directions are chaotic;

(2) Compared with the other two directions, the correlation dimension of the axial measurement points (Y-axis) of the main pipe is obviously smaller than that of the other two directions. It shows that the axial vibration of the pipeline has a smaller dimension chaotic attractor and requires fewer independent control variables to describe the dynamic system. This is mainly because the direction of the centrifugal force generated by the centrifugal pump of units is not in the axial direction of the main pipeline;

(3) At the same points, the $D_2$ of each point in condition 4 (No. 4 and 5 units in stable operation) is greater than in other conditions, while the corresponding $D_2$ in condition 3 (No. 4 unit in closing) is less. It indicates that the pipeline vibration is more complicated in the stable conditions of the two units and the complexity of the pipeline vibration is relatively weak in the closed condition. Unit operation increases the uncertainty of the pipeline vibration;

(4) In the same condition, the points near the units (point 1 and 5) and the bifurcated pipe (point 4) reach a relatively larger $D_2$, indicating that the vibration complexity of the pumping station pipe is greatly affected by the unit's vibration and the flow pattern stability.

To verify the validity of the above analysis results, the chaotic characteristics of the pumping station pipeline are further analyzed by using the largest Lyapunov exponent $\lambda_1$. According to the time delay $\tau$ and embedding dimension $m$, the small data sets calculate the largest Lyapunov exponent. Figure 8 is the $\lambda_1$ calculation diagram of typical points, and the value of the separation factor $y(i)$ tends to be stable after nearly linear growth. The linear slope is adjusted by the least square method, and the value is $\lambda_1$. The $\lambda_1$ of each point in different vibration directions are shown in Figure 9.

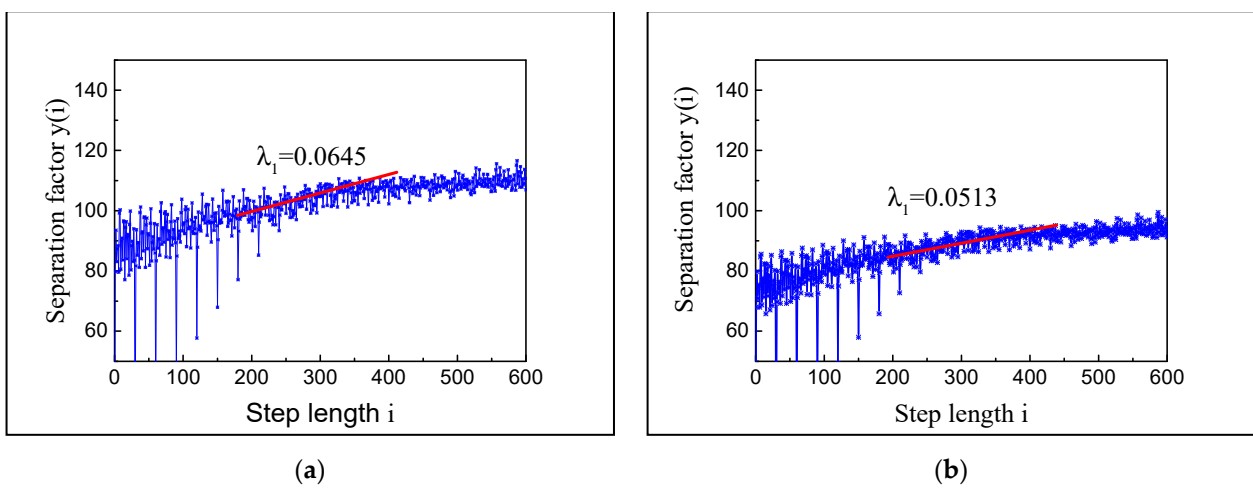

**Figure 8.** Calculation diagram of typical points (**a**) Point 3 Y-axis vibration under condition 1, and (**b**) Point 1 Z-axis vibration under condition 4.

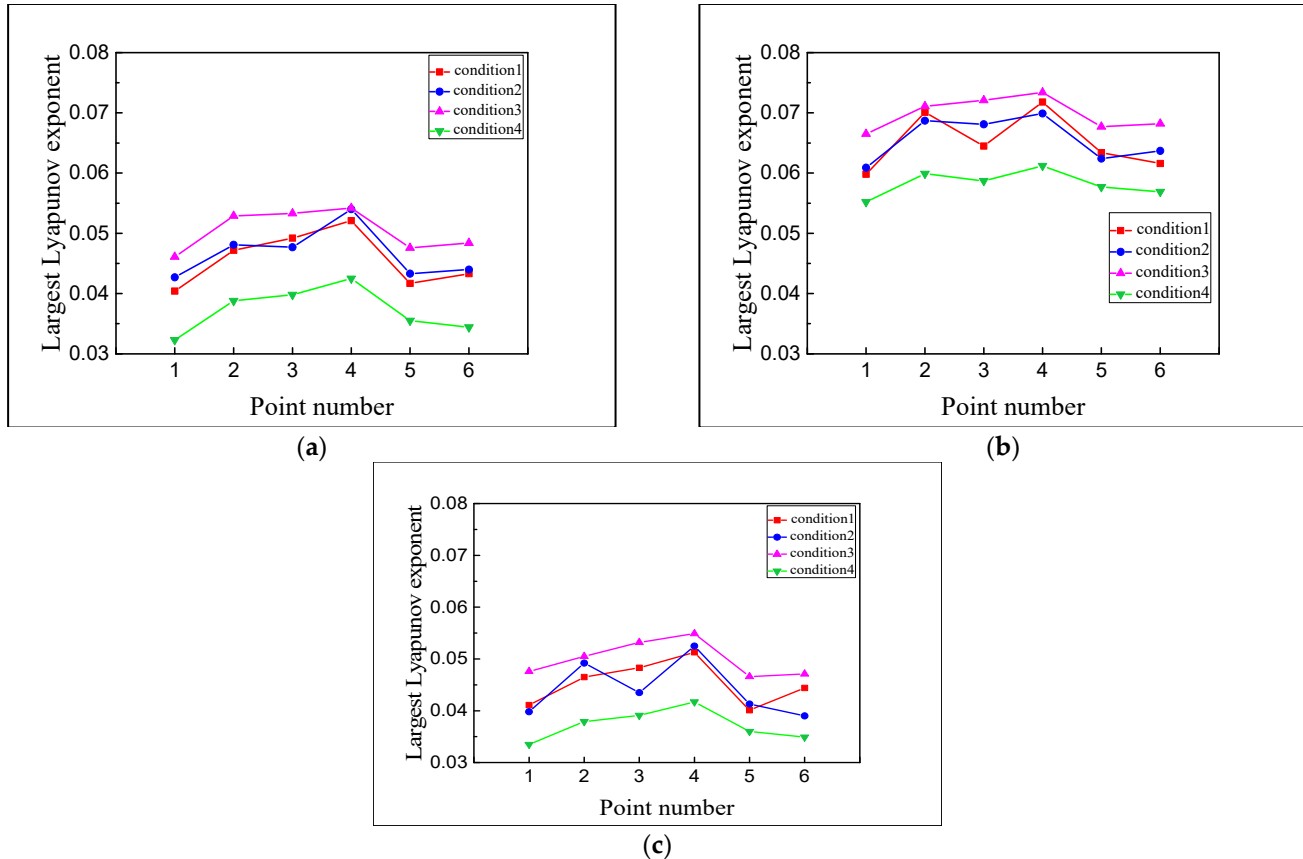

**Figure 9.** Largest Lyapunov exponent curves of points in different directions (**a**) X-axis points (**b**) Y-axis points, and (**c**) Z-axis points.

As shown in Figure 9:

(1)    The largest Lyapunov exponents $\lambda_1$ of different points are between 0.0323 and 0.0734, greater than 0. It shows that the measured vibration responses of pipelines have obvious chaotic characteristics. Also, the axial points $\lambda_1$ of the main pipeline (Y-axis) are obviously larger than the other two directions in the same condition, indicating the chaotic characteristics of the points separated from the influence of centrifugal force generated by pumping station units are more obvious;

(2)    The $\lambda_1$ of measuring points under condition 4 (No. 4 and 5 units in stable operation) are lower than in other conditions, while the $\lambda_1$ of condition 3 (No. 4 unit closing) is relatively larger. The largest Lyapunov exponent $\lambda_1$ decreases with the start-up of the two units, indicating that the units' operation weakens the chaotic characteristics of the pipeline vibration;

(3)    In the same condition, the $\lambda_1$ of the points near the units (point 1 and 5) are smaller. In contrast, while the $\lambda_1$ of the points at the pipeline's bifurcation (point 4) are greater than that of other measuring points, indicating that the sudden change of the flow state in the pipeline makes the vibration more chaotic, and the units' operation reduces the chaotic degree of vibration signals near the units.

The above analysis is complementary to the calculation results of the saturation correlation dimension $D_2$, which further confirms that the unit's operation and flow state changes greatly impact the chaotic characteristics of the pumping station pipeline.

## 4. The Analysis of Multi-Time-Scale Chaotic Characteristics Based on IVMD

The vibration characteristics of the pumping station pipeline are different from those of the general pipeline, which is mainly reflected in the influence of the pumping station unit on the vibration of the connecting pipeline. The vibration sources are primarily composed of low-frequency water pulsations caused by the pipeline flow and blade frequency, rotation frequency and frequency doubling produced by the unit's operation [23].

Taking the vibration response of the specific point (point 1 Z-axis vibration under condition 4) as an example, the spectrum analysis is shown in Figure 10. Concerning the author's previous article [23,24], 20, 40 and 60 Hz are the blade frequency, the rotation frequency and the frequency doubling, respectively, and 0.5 Hz is the low-frequency water pulsation. Spectrum analysis shows that the frequency band of the vibration excitation caused by water pulsation (0.5 Hz) is relatively wide. The wide-peak power spectrum is the typical characteristic of the chaotic system [25,26]. The pipeline vibration excitation produced by the unit's operation (20 Hz, 40 Hz, and 60 Hz) corresponds to the peak power spectrum and has high periodicity. Therefore, it is speculated that the chaotic characteristics of the pipeline are mainly caused by water pulsation, while the unit vibration masks the chaotic characteristics of the pump station pipeline.

The excitation components of different time scales must be effectively separated to clarify the vibration excitation with chaotic characteristics. As a new signal decomposition method, IVMD can adaptively decompose a signal into a series of IMFs with different scale characteristics. Therefore, the IVMD method is used to identify the vibration excitation that causes the chaotic characteristics of the pipeline.

The multi-time scale chaotic vibration response characteristics of the specific point (point 1 Z-axis vibration under condition 4) are analysed.

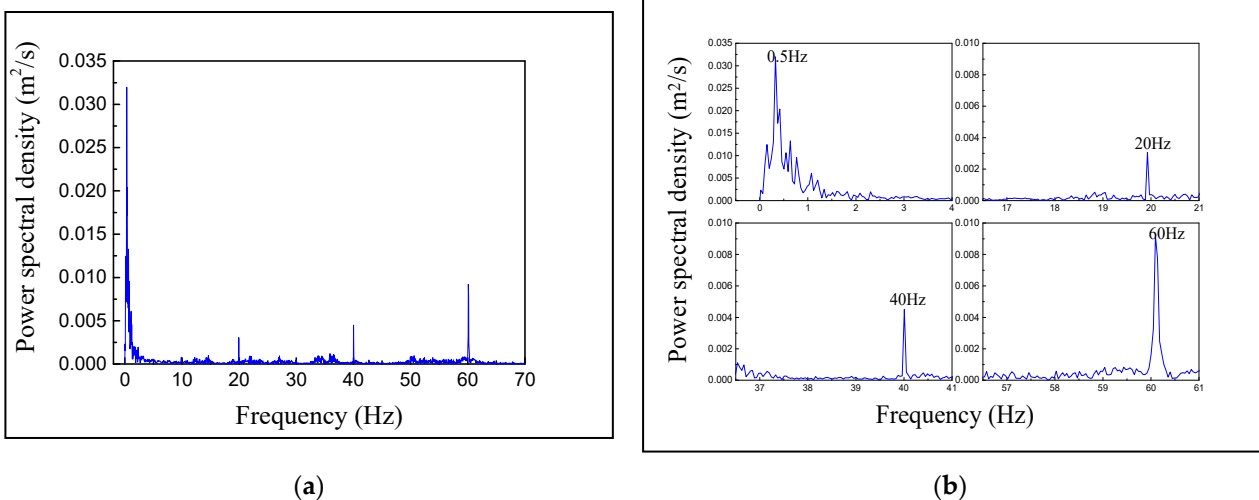

**Figure 10.** Spectrogram of point 1 Z-axis vibration response under condition 4 (**a**) Holistic drawing, and (**b**) Partial enlarged drawing.

The modal parameters *K* of IVMD are determined as 4 by the MI method. Four IMFs are obtained by the IVMD decomposition of point 1 Z-axis vibration response under condition 4. Figure 11 is the time history of decomposed IMFs.

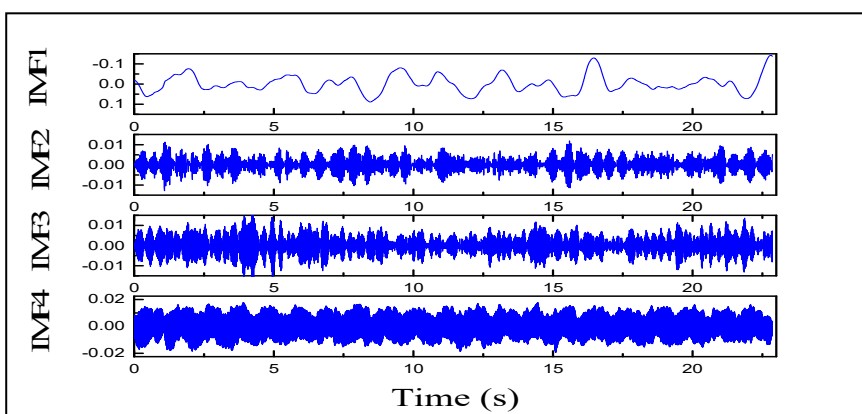

**Figure 11.** Time histories of IMFs decomposed by IVMD.

Mutual information value: IMF1 is 1.000, IMF2 is 0.025, IMF3 is 0.038, IMF4 is 0.0661. It can be seen that the normalized mutual information values of the IMFs are all above the threshold of 0.02, which meets the decomposition requirements. Figure 11 shows that IVMD can sequentially decompose the original vibrational response to obtain four IMFs with increasing frequency. The frequencies from IMF1 to IMF4 correspond to four major frequency bands in the original response spectrum: 0.5, 20, 40 and 60 Hz, respectively, and the decomposition effect is improved. Then the chaotic characteristics of the decomposed IMFs are analysed using the saturation correlation dimension and the largest Lyapunov exponent.

The calculation process of typical IMF chaotic eigenvalues is shown in Figure 12. From Figure 12a,b the saturation correlation dimension $D_2$ of IMF1 is 1.115, and the largest Lyapunov exponent $\lambda_1$ is 0.0774, indicating that IMF1 has prominent chaotic characteristics. The near-linear region of the $D_2$ logarithmic curve cannot be found from IMF2 to IMF4; these components have no chaotic characteristics. Due to space limitations, only the slope of the IMF2 double logarithmic curve is given in Figure 12c.

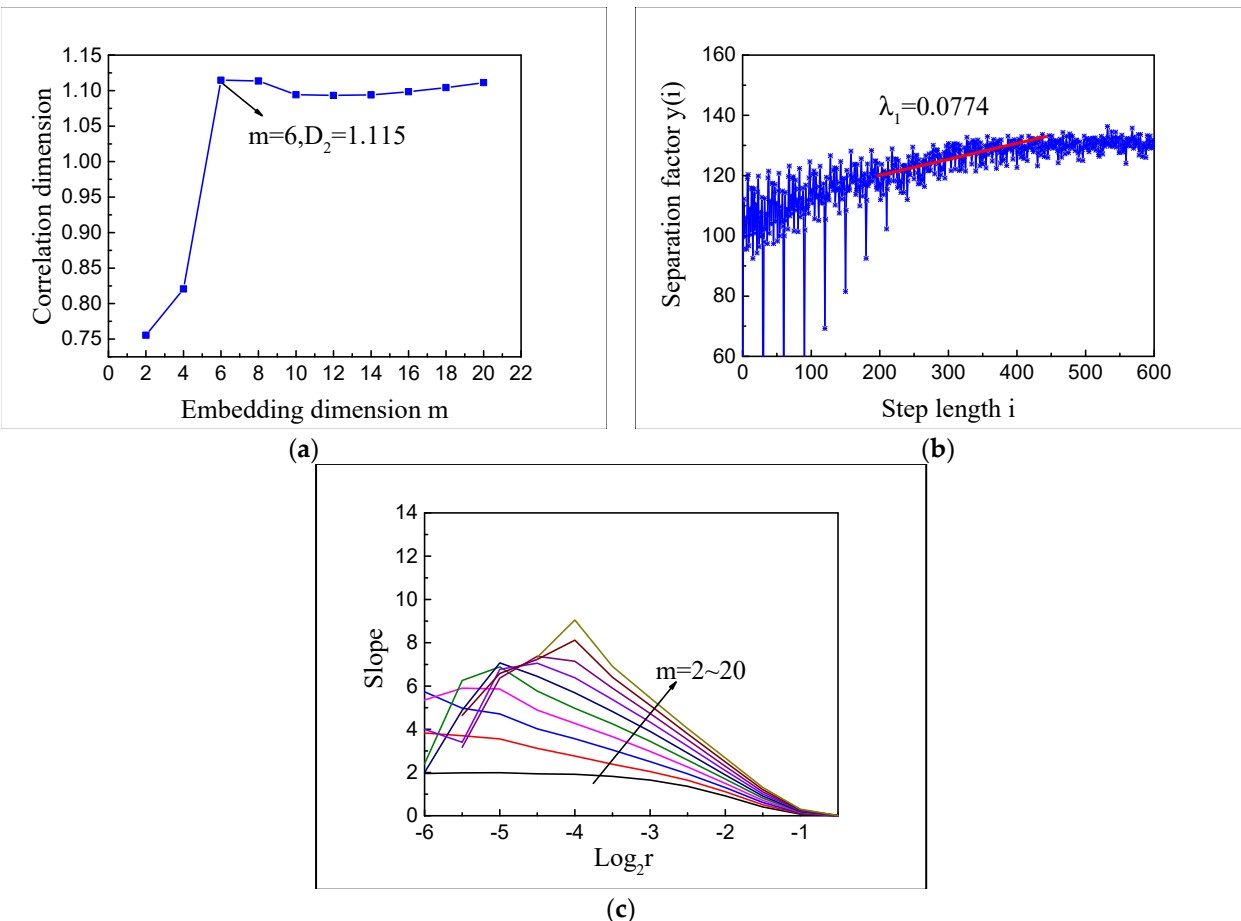

**Figure 12.** Calculation of chaotic eigenvalues of typical IMFs (**a**) Relation between $D_2$ and m of IMF1 (**b**) $\lambda_1$ Separation factor function of IMF1 (**c**) Slope of double logarithmic curve of IMF2.

By comparing the results of the chaotic eigenvalues of the IMFs with those of the vibrational response before decomposition, it can be concluded that:

(1) IMF1, which represents the water pulsation excitation, the saturated correlation dimension 1.115 is a fractal dimension, and the largest Lyapunov exponent is 0.0774 greater than zero, has prominent chaotic characteristics. IMF2 to IMF4, which represent the vibration excitation of the unit's operation, do not have any chaotic characteristics, indicating that the unit's operation cannot cause chaotic characteristics of the pumping station pipeline vibration;

(2) After eliminating the IMFs (IMF2 to IMF4) caused by the unit's operation with no chaotic characteristics, the saturation correlation dimension $D_2$ of the pipeline vibration response decreases from 4.985 to 1.115. At the same time the largest Lyapunov exponent increases from 0.0513 to 0.0774, that is, the complexity of the pipeline vibration decreases, and its chaotic characteristics are more evident. This shows that when the pumping station pipeline vibrates, the water pulsation excitation makes its vibration have obvious chaotic characteristics. In contrast, vibration excitation generated by the unit's operation masks the chaotic characteristics of the pumping station pipeline and increases the uncertainty of the pipeline vibration.

## 5. Conclusions

(1) Comparing the saturation correlation dimension $D_2$ among the vibration responses of the pumping station pipeline under different conditions, the $D_2$ of the measuring points are distributed in the range of 1.156–5.283, and all are fractions, which show that the vibration of the pumping station pipeline has chaotic characteristics. The

axial vibration of the pipeline presents a chaotic attractor with a lower dimension (1.156~2.569), and the vibration form is relatively simple. At the same time, the $D_2$ of conditions and points which are greatly affected by the unit's operation have a larger value (3.021~5.283), and the vibration form is more complex;

(2) The Lyapunov exponents $\lambda_1$ of measuring points under different conditions are between 0.0513 and 0.0774. With the opening of two units, the largest Lyapunov exponent $\lambda_1$ decreases accordingly, suggesting that the unit's operation weakens the chaotic characteristics of the pipeline vibration. The $\lambda_1$ of points at the bifurcation are larger than those of other points under the same condition. The chaotic characteristics of the vibration at the bifurcation are enhanced by the sudden expansion of the pipe diameter at the bifurcation and the impact of water heads at different flow velocities;

(3) After the IVMD decomposition of the vibration response of specific points under the unit's operation conditions, the chaotic characteristics of the IMFs are analysed. The results show that the saturation correlation dimension $D_2$ of IMF1 representing water pulsation excitation in the pipeline is 1.115, and the largest Lyapunov exponent is 0.0774. The IMF2 to IMF4 representing the blade frequency, the rotation frequency, and the frequency doubling vibration excitation generated by the unit's operation do not have chaotic characteristics. It indicates that the chaotic character of the pumping station pipeline is mainly caused by water pulsation in the pipeline, and the vibration caused by the unit masks the chaotic characteristic of the pipeline, which makes the pipeline vibration system more complex.

In this paper, the chaotic characteristics of the vibration system of the pumping station pipeline are shown by the analysis of the measured vibration responses, and the chaotic excitation is found by combination with IVMD, which provides a theoretical basis for the complete description of the vibration characteristics of the pumping station pipeline. A new way of chaotic characteristics analysis based on IVMD decomposition is also proposed.

**Author Contributions:** Conceptualization, L.J. and Z.M.; methodology, L.J.; software, J.Z. and L.W.; validation, L.J. and J.Z.; formal analysis, J.Z. and M.C.; investigation, L.J.; resources, Z.M.; data curation, J.Z.; writing—original draft preparation, L.J.; writing—review and editing, M.Y.A.K.; visualization, L.J.; supervision, Z.M.; funding acquisition, J.Z. All authors have read and agreed to the published version of the manuscript.

**Funding:** This research was funded by Program for Science & Water conservancy science and technology innovation project in GuangDong Province, grant number 2020-18.

**Institutional Review Board Statement:** Not applicable.

**Informed Consent Statement:** Not applicable.

**Data Availability Statement:** The data provided in this study are available from the corresponding author.

**Conflicts of Interest:** The authors declare no conflict of interest.

## Abbreviations

IVMD    improved variational mode decomposition
IMF      intrinsic mode function

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
