# Peer review of "Chaotic Characteristic Analysis of Vibration Response of Pumping Station Pipeline Using Improved Variational Mode Decomposition Method"

_applsci, doi:10.3390/app11198864_

Round 1

Reviewer 1 Report

see attached

Reviewer 2 Report

Brief Description of the Work

The objective of the work is to analyse the chaotic characteristics of vibration responses of pumping station pipeline in an irrigation area, under four working conditions, by using the saturation correlation dimension and the largest Lyapunov exponent. The Improved Variational Mode Decomposition (IVMD) method is used to identify the vibration excitation that causes the chaotic characteristics of the pipeline. The chaotic characteristics of the intrinsic mode functions are analysed after the IVMD decomposition of vibration response of typical points under unit operation conditions,

Main Obtained Results

1) The axial vibration of the pipeline presents a chaotic attractor (with dimension 1.156~2.569);

2) The unit operation weakens the chaotic characteristics of pipeline vibration;

3) The Lyapunov exponents of points at the bifurcation are larger than those of other points under the same condition;

4) The chaotic characteristics of the vibration at the bifurcation are enhanced by the sudden expansion of pipe diameter at the bifurcation and the impact of water heads at different flow velocities;

5) The chaotic character of the pumping station pipeline is mainly caused by water pulsation in the pipeline, and the vibration caused by the unit masks the chaotic characteristic of the pipeline.

General Considerations (GC)

GC1) English needs to be widely revised.

GC2) The writing of the formulas and the text-layout must be improved. The standard template of the journal MDPI Applied Sciences should be used.

GC3) The present work is not sufficiently framed in the context of the works that have recently appeared in the literature in this matter. The authors should compare their method of analysis with those currently proposed in the literature, by showing the advantages and limitations of both.

GC4) There are some points that are not clear to me. The following questions aim to identify some of them.

Specific Questions (SQ)

SQ1) In the authors' work the vibration sources are mainly composed of low-frequency water pulsations caused by pipeline flow and blade frequency, rotation frequency and frequency doubling produced by unit operation. However, in general, vibrating motions are classified as oscillating, reciprocating, or periodic. Vibration can also be either harmonic or random (harmonic vibration occurs when a vibration's frequency and magnitude are constant. A vibration is random when the frequency and magnitude vary with time). However, the analysis conducted by the authors was not carried out in terms of this (standard) classification of vibrations. The authors are kindly invited to elucidate this aspect.

SQ2) Variable “r” in equation (2) at page 3 (lines 97 and 98) is not specified. Please define this variable by explaining its physical meaning.

SQ3) The authors found that the axial vibration of the pipeline presents a chaotic attractor. Attractors typically have a fractal structure, and the fractal dimension can be calculated. The dimension of the attractor found by the authors (1.156~2.569) coincides with the fractal dimension? Please clarify the link between the correlation dimension D2 estimated by the authors though equation (3) (which is calculated as the log of the correlation function defined in equation (1) and log of "r" - not well defined by the authors, see the previous question) and the fractal dimension (which is calculated as the log of the number of pieces divided by the log of the magnification factor).

SQ4) It is known that there exist nonlinear effects introduced by motion-limiting constraints. In particular, experiments indicate that, with increasing flow velocity, beyond the Hopf bifurcation there are regions of period-doubling and chaos. Did the authors see the Hopt bifurcation in their pipeline system? Please discuss this point.

SQ5) The authors found that the vibration of the pumping station pipeline has chaotic characteristics. However, the routes to chaos from limit cycle has not been described. More specifically, please specify if:

a) chaos is due to a cascade of subharmonic bifurcation (period doubling cascade);

b) the system shows intermittency;

c) the authors have observed homoclinic orbits in a Poincaré map associated with a phase space flow as precursors of the chaotic motion. Chaos is due to the production of a pair of heteroclinic intersections.

If the authors found the route transition a), did they find the Feigenbaum constant δ = 4. 715?

SQ6) Please specify the nature of the material of the pipelines. As known, some materials reduce vibration stress and the authors findings reported above. The most common type of bellows is made of metal (most commonly stainless steel), plastic (such as PTFE), fabric (such as glass fibre), or an elastomer such as rubber.

SQ7) A common approach to vibration suppression is adding damping. Vibration damping dissipates some of the vibration energy by transforming it to heat. Did the authors investigated how the chaotic characteristics of vibration responses of their pumping station pipeline are modified in presence of the vibration damping or of a passive vibration absorber in the pipeline system ? Note that this is a very important aspect in designing a pipeline system and the analysis of (passive) vibration damping is generally included in the study.

SQ8) (It is not mandatory to answer this question).

Fluid flows in piping passing the entry to a closed end side-branch, may generate vortices which coincide with strong acoustic resonances in the side-branch and result in pulsations being generated which propagate both upstream and downstream within the main line. Did appear this phenomenon in the authors’ experiments? If yes, can the authors describe briefly, in a qualitative way, the influence that pulsations can have on the chaotic characteristics of the vibration responses of the pumping station pipeline?

 Conclusions

In my opinion, the work cannot be published in the present form. Authors are encouraged to answer the questions raised in the Section "Specific Questions" and to take into account the above suggestions (ref. to the Section "General Considerations").

Round 2

Reviewer 1 Report

the paper is amended accordingly

Reviewer 2 Report

The authors answered point by point, and satisfactorily, to the questions raised in my report. In my opinion, the present version deserves to be published.